# Locoregional Treatment in Intrahepatic Cholangiocarcinoma: Which Treatment for Which Patient?

**DOI:** 10.3390/cancers15174217

**Published:** 2023-08-23

**Authors:** Héloïse Bourien, Chiara Carlotta Pircher, Boris Guiu, Angela Lamarca, Juan W Valle, Monica Niger, Julien Edeline

**Affiliations:** 1Medical Oncology Department, Centre Eugène Marquis, 35000 Rennes, France; j.edeline@rennes.unicancer.fr; 2Medical Oncology Department, Fondazione IRCCS Istituto Nazionale Tumori, 20133 Milano, Italy; chiara.pircher@istitutotumori.mi.it (C.C.P.); monica.niger@istitutotumori.mi.it (M.N.); 3Interventional Radiology Department, CHU de Montpellier, 34090 Montpellier, France; b-guiu@chu-montpellier.fr; 4Oncology Department, Fundación Jiménez Díaz University Hospital, 28022 Madrid, Spain; angela.lamarca@nhs.net; 5Medical Oncology Department, Division of Cancer Sciences, University of Manchester, Manchester M13 9PL, UK; juan.valle@nhs.net; 6The Christie NHS Foundation Trust, Manchester M20 4BX, UK

**Keywords:** cholangiocarcinoma, locoregional treatment

## Abstract

**Simple Summary:**

Due to the rarity of the entity of cholangiocarcinoma, there is a lack of randomized clinical trials which can compared modalities of treatment for unresectable intra-hepatic cholangiocarcinoma (iCC). In this review, we proposed to summarize current evidence regarding all the modalities of loco-regional treatment in iCC in order to help clinicians in their decision-making.

**Abstract:**

For unresectable intrahepatic cholangiocarcinoma (iCC), different locoregional treatments (LRT) could be proposed to patients, including radiofrequency ablation (RFA) and microwave ablation (MWA), external beam radiotherapy (EBRT) or transarterial treatments, depending on patient and tumor characteristics and local expertise. These different techniques of LRT have not been compared in a randomized clinical trial; most of the relevant studies are retrospective and not comparative. The aim of this narrative review is to help clinicians in their everyday practice discuss the pros and cons of each LRT, depending on the individual characteristics of their patients.

## 1. Introduction

Biliary tract cancers (BTC) are heterogenous entities comprising intrahepatic cholangiocarcinoma (iCC), perihilar cholangiocarcinoma, distal cholangiocarcinoma, gallbladder cancers and sometimes ampullary cancer. Most patients are diagnosed at an advanced (locally advanced or metastatic) stage. Currently, for these patients, a combination of gemcitabine and cisplatin is recommended in the first-line setting, based on the results of the Advanced Biliary Tract Cancer (ABC)-02 and BT22 trials [1,2]. Recently, durvalumab showed a significant, albeit modest, improvement in overall survival (OS) and was recently granted approval by the FDA [3].

For liver-only iCC, surgeons have to evaluate whether a complete resection, R0, could be performed as surgery is the only potential curative treatment for these patients. Nevertheless, outcomes remain poor, the median OS after curative-intent surgery is about 30 months, and only about a third of patients experience long-term, relapse-free survival (RFS) [4,5,6]. 

For unresectable liver-only iCC, or for patients not suitable for surgery, radiofrequency ablation (RFA), microwave ablation (MWA) and other locoregional treatments (LRT) like external beam radiotherapy (EBRT) or various transarterial treatments, including transarterial chemoembolization (TACE), radioembolization (also known as selective internal radiation therapy, or SIRT) or hepatic arterial infusion chemotherapy (HAIC), have also been investigated for unresectable non-metastatic iCC.

The aim of this paper is to summarize current evidence regarding all the modalities of LRT for iCC in order to help clinicians in their decision making. First of all, we describe the literature available on the different modalities of LRT in iCC. We based this on a recent systematic review of the literature [7], adding interpretations and a discussion about each modality, specifically a discussion regarding the best patient profile for each modality. Secondly, we summarize relevant criteria needed to decide which treatment would be the most appropriate for each patient in a personalized approach.

## 2. Narrative Review about Current Data Regarding LRT of iCC

### 2.1. Radiofrequency Ablation (RFA) and Microwave Ablation (MWA)

Many studies have reported results of RFA and MWA in iCC (Table 1). There is no randomized trial evaluating ablative therapy; only prospective cohorts or retrospective studies were published. Studies are heterogenous, which leads to difficulties in cross-trial comparison.

Overall, these studies suggest adequate local control following ablation of iCC. In a recent review, the reported response rate was 93.9% [7].

One of the most important characteristics for using RFA or MWA is the tumor size. In retrospective studies, the tumor size ranged from 0.7 to 10 cm, but the better control of tumors was observed for tumors < 5 cm. For Brandi et al. [8], a tumor size less than 2 cm was an independent factor for improved local tumor progression-free survival. For the cohort of Díaz-González Á [9], the median time to recurrence was significantly lower for a tumor smaller than 2 cm. This correlates well with data from hepatocellular carcinoma (HCC), where RFA is considered equivalent to surgery for tumors up to 2 cm and acceptable for tumors up to 3 cm. Moreover, the ALBI grade and number were independent prognostic factors for ablative therapies in terms of OS [10,11,12]. There is no impact of age on progression-free survival (PFS) or OS after RFA and MWA [13], suggesting that they could be proposed for older patients.

The main reported complications include ascites or pleural effusion, liver abscess, portal vein thrombosis, jaundice and hepatic failure; however, the rate of severe complications is low overall. For Xu et al. [10], the rate of complications of MWA was 5.3%, half that seen with surgery (13.8%). The overall procedure-associated major complication rates range from 2.8% to 5.5% [10,12,13,14]. Kim et al. [15] reported only one complication (liver abscess) out of 13 patients, for a lesion of 7 cm in diameter treated by RFA.

To conclude, RFA and MWA are effective and safe treatments for iCC. They could be proposed as curative-intent treatments in patients deemed unfit for major surgery who have a limited number of lesions (up to 3), lesions under 3 cm, and who have good liver function according to the ALBI score, with or without the presence of cirrhosis (Figure 1A). 

**Table 1 cancers-15-04217-t001:** Studies that evaluated RFA or MWA in iCC.

Authors	Methods	Retrospective or Prospective Study	Patients (n)	Tumor (n)	Median Tumor Size (cm)	Number of Lesions	Extrahepatic Disease Patients (n) or % of Patients	Efficacy: Local Tumor Progression	Grade 3–5 Treatment Related Toxicities
		One arm cohort							
Butros [16]	RFA	Retrospective study	7	9	2.4 (1.3–3.3)	1–2		1/9 (11%)	No major complication
Fu [17]	RFA	Retrospective study	17	26	4.4 (2.1–6.9)	1–5	7%	3/17 (17.6%)	1 major compli- cation occurred (3.6%, 1 of 28 sessions) (pleural effusion)
Kim [15]	RFA	Retrospective study	13	17	0.8–8	1–2		6/17 (35.3%)	1 patient died following liver abscess
Kim [18]	RFA	Retrospective study	20	29	1.5 (0.7–4.4)			6/29 (20.7%)	2 major complications occurred (7%) (liver abscess, bile duct stenosis)
Carrafiello [19]	RFA	Retrospective study	6	6	3.45 (1.0–5.8)			3/6 (50%)	No major complication
Chiou [20]	RFA	Prospective cohort	10	10	1.9–6.8	1		2/10 (20%)	No major complication
Haidu [21]	RFA	Retrospective study	11	36	3 (0.5–10)		3%	3/36 (8%)	Major complication rate: 13% (3/23) (bleeding, pseudoaneurysm, pulmonary embolism)
Brandi [8]	RFA	Retrospective observational cohort study	29	117	1.7 (0.5–4.8)			Local tumor progression-free survival: 9.27 months (7.34–11.15)	Major complication rate: 7% (8/117) (liver abscess, pleural effusion, biloma, intrahepatic hematoma)
Díaz-González [9]	MWA (microwave) and RFA	Retrospective analysis	27		2.1	1–2	0	21/27 (77.8%)	Data not known
Ni [11]	MWA	Retrospective study	78	106	3.1 (0.8–5.0)	1–3	0		3 patients (3.8%) had major complication (liver abscess, pleural effusion)
Takahashi [22]	6 MWA and 44 RFA	Retrospective review	20	50	1.8 (0.5–4.7)			11/50 tumor Or 5/20 patients	No major complication
Zhang [13]	MWA	Retrospective study	107	171	<5	1–3			3 patients (2.8%) presented major complication (pleural effusion, liver abscess)
Xu [14]	MWA and RFA	Retrospective study	18	25	2.8 (0.7–6.9)	1–4	0	12/18	1 patient presented major complication (fever)
Ge [23]	MWA	Retrospective study	92 (compared to 183 TACE)		3.3–8.1				Data not known
Giorgio [24]	MWA versus RFA	Retrospective study	71 36 RFA 35 MWA	98	3.6 (2.2–7.2)		0		No major complication
Xu [10]	MWA versus surgery	Retrospective study	121 56 MWA 65 surgery		2.7 (0.8–5.0)		11/56 (19.6%)		MWA: rate of major complication was 5.6% (3/56) (hepatic failure, ascites, liver absces)
Zhang [12]	Ablation vs. surgery	Retrospective study	32 surgery 77 ablation		<5		0		Major complication rate: 3.9% (3/77) (hepatic failure, liver abscesses)

### 2.2. External Beam Radiotherapy (EBRT)

Several teams evaluated radiotherapy for the management of iCC (Table 2). Radiotherapy could be proposed in curative or in palliative intent. There is a high heterogeneity concerning dose, schedules or techniques (some used proton therapy, while others used photon therapy or SBRT). Depending on individual studies, the gross tumor volume (GTV) was not the same for liver tumor or regional disease. Moreover, some groups proposed chemotherapy concomitantly [25,26] or sequentially [26,27] with different drugs (5-fluorouracil, gemcitabine, cisplatin, and doxorubicin [26]; erlotinib [28]; or S-1 [29]) administered via hepatic arterial infusion or systemically. Before receiving radiotherapy, patients could have received other treatment, such as surgery, chemotherapy, transarterial treatment, RFA and MWA, or transplantation [30]. One other difficulty in interpreting the data in the literature is the fact that, in some series, the patients included suffered from HCC or iCC, without distinguishing the two entities for toxicity or clinical outcome analyses.

Overall, the literature shows large variations in local control rates and PFS rates, which is probably related to different selections of the population, tumor characteristics, and the technique applied. In the review by Edeline et al. [7], the mean 2-year local control is 69.1% (95% CI: 48.1–90.2).

In the different series of radiation therapy for iCC treatment, grade 3 and higher adverse events were observed in around 10% of patients [25,31,32,33] but can be observed in up to 20% [28]; however, in some series, no grade 3 toxicities were observed [29]. The more frequent grade 3 or higher radiation-related toxicities experienced by patients were cytopenia, gastro-intestinal and hepatic toxicity. Liver failure may occur due to radiation-induced liver disease (RILD) and could be fatal. Depending on the series, RILD was observed in 0 to 7% of patients, but it is rarely fatal. Nevertheless, in a phase I/II study, 2 out of 26 patients (7%) died from liver dysfunction—one probably due to RILD [28]. The likelihood of liver dysfunction and RILD correlates with the volume of spared liver. The addition of chemotherapy to radiotherapy does not seem to increase grade 3 toxicities [26]. The dose of radiation delivered to the hepatobiliary tree seems to be a predictive factor of complication. Shen et al. [34] did not observe toxicities greater than grade 1 for patients with tumors smaller than 5 cm.

To conclude, external beam radiation therapy seems to be a possible treatment for iCC treatment, achieving a satisfactory local control rate for patients with good liver function (Child A or B) and with a tumor size under 5 or 7 cm. EBRT could be proposed for tumors with vascular contact or tumors technically not accessible or too large for RFA or MWA. On one hand, a tumor that received a dose was one of the most important predictive factors of local control; on the other hand, the liver volume spared was correlated with liver dysfunction. Patients need to be carefully selected to obtain the best control rate with the lowest RILD rate. To avoid liver dysfunction and have the best control rate, a tumor size under 7 cm seem to be more suitable [33]. Radiation therapy could be discussed in patients not suitable for surgery, in patients for which RFA or MWA are technically not possible, or in palliative intent (Figure 1B).

### 2.3. Intra-Arterial Treatment (IAT)

Different intra-arterial treatments could be proposed for iCC, including SIRT, TACE and HAIC [7].

#### 2.3.1. Yttrium-90 Microsphere Selective Internal Radiation Therapy 

For cholangiocarcinoma, SIRT indications are not well defined and depend on individual treating centers and their practice. Several groups have reported their retrospective experience of SIRT for the management of patients with cholangiocarcinoma (Table 3). As with radiotherapy, SIRT may be performed alone or in combination with chemotherapy. In some retrospective series, SIRT has been used as consolidation in patients with no disease progression after first-line chemotherapy and without extrahepatic disease [35]. Only one clinical trial evaluated SIRT with concomitant chemotherapy (n = 41) [36]. Contrary to radiotherapy, chemotherapy used concomitantly with SIRT seems to be standardized and defined by the doublet gemcitabine and cisplatin standard of care regimen [36]. SIRT may be used for downstaging, aiming to render patients suitable for surgery with curative intent or for patients who are in a palliative setting.

There is a large variation in reported efficacy results, probably due to the heterogeneity of the population included. Results in the first-line setting seem promising in the single-arm MISPHEC clinical trial [36] evaluating the activity of gemcitabine-cisplatin with SIRT for patients affected by unresectable iCC with no or limited extrahepatic disease. This strategy permitted a high disease control rate of 98% (95% CI, 80–99%) at 3 months. Of the 41 patients initially considered unresectable, 9 patients (22%) could be downstaged to surgical intervention. Median PFS was 14 months (95% CI, 8–17 months), and median OS was 22 months (95% CI, 14–52 months).

**Table 2 cancers-15-04217-t002:** Studies that evaluated radiotherapy in iCC.

Authors	Retrospective or Prospective Study	Treatment Dose	Patients (n)	Extrahepatic Disease Patients (n) or % of Patients	Median Tumor Size (cm)	Efficacy	Grade 3–5 Treatment-Related Toxicities
Non-comparative arm							
Shimizu [25]	Retrospective study	Range: 46.6 Gy in 12 fractions to 74.0 Gy in 37 fractions 16 patients received concomitant CT	37		5.7 (1.5–14)	The 1-year local control rate: 97.3% (95% CI: 92.0–100%)	3 patients experienced grade 3 biliary tract infections
Smart [31]	Retrospective study	Median dose: 58.05 Gy (37.5–67.5)	66	23/66 patients	5.6 (2.5–16)	Disease recurrence: 42/66 (74%) Local failure: 5/66	11% (7/66) patients had grade 3 and 4 toxicities (thrombocytopenia, neutropenia, nausea, anorexia, abdominal pain, dehydration, fever, RILD)
Kozak [27]	Retrospective study	Median dose: 40 Gy (26–50) Median fractions: 5 (1–5)	40		4.2 (1.0–12.5)	Local failure: 12/40	16 patients (40%) experienced grade 3 toxicity (abdominal pain, infection, biliary complication, liver abscess, cholecystitis, elevated liver enzymes)
Kasuya [37]	Retrospective study	Carbon-ion radiotherapy Most commonly prescribed dose: 76 Gy in 20 fractions	56		3.7 (1.5–11)	The 1-year local control rate: 79.4% (IC 95%: 62.7–89.2)	1 patient died of liver failure, 3 patients had liver dysfunction, and 1 patient presented a bile duct stenosis
Shen [34]	Retrospective study	Median dose: 45 Gy (36–54)	28			The 1-year PFS rate: 50%	28 patients had at least one grade 3 toxicity (gastrointestinal ulcers elevated liver enzyme, hematological toxicity).
Cho [26]	Retrospective study	Concomitantly with chemotherapy: For IMRT: 45 Gy to the PTV For 3D-CRT: 45 Gy in 25 fractions to the PTV	64 patients concomitantly with chemotherapy, and 56 underwent surgery	0/120 patients		The 3-year locoregional failure-free survival: 50% for patients who underwent surgery after radio–chemotherapy	7.8% (5/64) patients had grade 3 toxicities (nausea, vomiting, epigastric pain, gastric bleeding)
Weiner [28]	Prospective trial	Median dose: 55 Gy (40–55)	26 patients but 12 HCC, 12 iCC and 2 mixed		5.5 (1.6–12.3)	The 1-year local control rate: 91% 1-year PFS: 50% (95% IC: 29–69%)	11 patients presented ≥ grade 3 toxicities (hematological toxicity, hepatic failure, abdominal pain, elevated liver enzymes, ascites, vomiting and skin fibrosis)
Hong [30]	Prospective trial		83 patients: 44 HCC 39 iCC	0	5.7 (1.9–12.0) For iCC: 6 (2.2–10.9)		4 patients (4.8%) presented grade 3 toxicity (thrombocytopenia, liver failure, ascites, gastric ulcer and elevated bilirubin)
Tao [38]	Retrospective study	58.05 Gy (35–100)	79		7.9 (2.2–17)	The 1-year local control rate: 81%	4 patients had major complication (cholangitis, gastric bleeding)
Ohkawa [29]	Retrospective study	Median total proton dose: 72.6 Gy in 22 fractions for intrahepatic region	20 12 curative 8 palliative (4 stage IV and 4 stage IIIC for which the irradiation was not sufficient due to a too-wide tumor size	4/20 patients	5.0 (1.5–14)	The 1-year local control rate: 88% for the curative group	5% (1/20) patients had grade 3 toxicities (bone marrow suppression)
Jung [32]	Retrospective study	45 Gy in 3 fractions (range: 15 to 60 Gy in 1–5 fractions) SBRT alone or EBRT and SBRT	58			The 1-year local control rate: 85%	6 patients (10%) experienced a toxicity ≥ grade 3 (cholangitis and bile duct stenosis, gastric perforation).
Kim [39]	Retrospective study	In association with chemotherapy: 44 Gy (25–60): 5 fractions of 2–3 Gy	92:25 in the arm chemo-radiation		7.6 ± 3.9	Disease control rate: 56%	Grade 3 neutropenia occurred in 3/25 patients (12%) Grade 3 thrombopenia occurred in 5 (20%) 6/25 (24%) patients had > grade 3 toxicities.
Ibarra [40]	Retrospective study	iCC: 30 Gy (22–50) 1–10 fractions	32:21 HCC 11 iCC	45.5%		The 1-year disease-free local progression: 50%	9% (3/32) patients had grade 3 or 4 toxicities
Tse [33]	Prospective trial	36 Gy (24–54)	41:31 HCC and 10 iCC	2/10 iCC	Tumor volume: 172 cm^3^ (10–465)	The 1-year local control rate: 65% (95% IC 44–79%)	18 events of grade 3 or 4 toxicities were observed (liver toxicity and nausea)
Yi [41]	Retrospective study	Chemoradiation	176	0		Response rate: 19.8%	Grade 3 thrombocytopenia occurred in 10.4% of patients

**Table 3 cancers-15-04217-t003:** Studies that evaluated SIRT in iCC.

Authors	Retrospective or Prospective Study	Patients (n)	Localisation	Patients with Extrahepatic Disease n (%)	Previous Treatment	Mean Activity (GBq)	Median Tumor size (cm)	Efficacy mOS or meanOS from the 1st RE	Grade 3–5 Treatment-Related Toxicities
Helmberger [42]	Prospective observational study	1050 in the whole cohort 120 iCC		36/120 (30%)	ICC patients: Chemotherapy: 39.2% received combined regimens based on gemcitabine Locoregional treatments: 34.2% (surgery for 26.7%)			mOS: 14.7 months (95% CI: 10.9–17.9)	Less than 2.5% patients presented grade 3–4 toxicities (gastritis, gastrointestinal ulcerations, radiation cholecystitis and REILD)
Azar [43]	Retrospective review	96 in whole cohort and 22 iCC	Bilobar: 63.6% Unilobar: 35.4%	2/22 (9.1%)	16/22 (72.7%): Surgery: 8/22 (36.4%) Radiotherapy: 5/22 (22.7%) Chemotherapy: 12/22 (54.5%) Locoregional treatment: 1/22 (4.5%)	1.5 (0.5–2.8)			Data not known
Bargellini [44]	Retrospective study SirSphere	81 in whole cohort: 35 (42.2%) in group A: first-line treatment at first diagnosis or at recurrence after surgery 19 (23.5%) in group B: SIRT as consolidation treatment after radio-logical disease control following first-line chemotherapy 27 (33.3%) in group C: SIRT because of tumor progression after first-line chemotherapy	Bilobar: 49.4%	8/81 (10%)	Surgery: 32/81 (39.5%)	1.46 ± 0.49	59.8 ± 32.5	mOS: 14.5 months (11.1–16.9)	No toxicity grade ≥ 3 was recorded
Buettner [45]	Retrospective study	115 92: SIR-Sphere 22: resin microsphere 1: with both	Bilobar: 72%	27/115 (24%)	Chemotherapy: 91/115 (79%)	Median administered activity measurements were 1.6 GBq (IQR (interquartile range), 1.3–1.9 GBq) for patients who received resin microspheres and 2.6 GBq (IQR, 1.5–3.8 GBq; *p* = 0.0017) for patients who received glass microspheres.	7.2 (5.4–10.0)	Median OS after treatment was 11 months (95% CI: 8–13)	4 patients experienced grade 3 toxicity (4%) (REILD)
Filippi [46]	Retrospective study	20 SIR-Sphere		8/20 (40%)	Chemotherapy: 11 patients Surgery (liver resection): 8 patients Ablation: 1 RT on metastatic site: 1 patient	1.6 ± 0.4 GBq		meanOS 12.5 ± 1.5 months	Data not known
Köhler [47]	Retrospective study	46 SIR-Sphere	Bilobar: 63%	14/46 (30.4%)	30/46 (65.2%) Chemotherapy: 28 patients Immunotherapy: 1 Radiotherapy: 4 Liver resection: 9 TACE: 1	Median: 1.74 (0.51–3.26)		mOS: 9.5 months (95% CI: 6.1–12.9)	Data not known
Edeline [36]	Phase 2 clinical trial SIRT in association with chemotherapy (GEMCIS) in 1st line	41	Unifocal: 34%	7/41 (17%) (lung metastasis ≤ 1 cm)	Resection: 5	The median dose delivered to the tumor was 317 Gy (range: 64–1673 Gy)		mOS: 22 months (95% CI: 14–52)	29 patients (71%) experienced grade 3 or 4 toxicities (gastrointestinal, hematological, hepatobiliary and general toxicities).
White [48]	Prospective single-arm observational	61 SIR-sphere (74%) and therasphere (26%)	Bilobar: 64%	22/61 (36%)	Chemotherapy: 56/61 (92%)			mOS: 8.7 months (95% CI: 5.3–12.1)	7 events of grade 3–4 toxicities (fatigue, fever and perturbation of liver function)
Bourien [49]	Retrospective study	64	Bilobar: 56%	10/64 (16%)	Resection: 15/64 (23%) Chemotherapy: 27/64 (42%)	2.5 (0.6–7.7)	7.7 (1.4–18.2)	16.4 months (95% CI: 7.8–25.0)	10 patients (16%) experienced grade 3 fatigue; 6 patients (9%) experienced grade 3 liver pain; 2 patients (3%) had grade 3 nausea; and 2 patients (3%) had hepatic failure. No grade 4 toxicity was reported
Gangi [50]	Retrospective study	85	Bilobar: 36.5% Solitary tumor: 61.2%	36/85 (42.4%)	Chemotherapy: 61/85 (71.8%) Liver resection: 14/85 (16.5%) Radiotherapy: 4/85 (4.7%)	Median delivered dose was 136.0 Gy		12.0 months (95% CI: 8.0–15.2)	1 patient developed a grade 3 toxicity (liver abscess)
Shaker [51]	Retrospective study	17 9 SIR-sphere 8 therasphere		5/17 (29.4%)	Chemotherapy: 5/17 patients	The thera-Sphere and SIR-Sphere groups were 158.2 ± 128.1 Gy and 34.5 ± 16.3 Gy, respectively	7.4 cm ± 3.3	mOS from the diagnosis 33.6 months (95% CI: 4–64.8)	Data not known
Reimer [52]	Retrospective study	21	Bilobar: 19% Solitary: 57%	3/21 (14%)	0			Median survival was 15 months	One of the patients had an ulceration of gastric mucosa.
Akinwande [53]	Retrospective study	25 SIRT 15 TACE		11/25 (44%)	Chemotherapy: 16/25 (64%) Surgery/ablation: 5/25 (20%)	1.56 GBq (0.41–5.31)			TACE: 3/33 (9%) grade 3 or more treatment-related toxicities (grade 3 or more treatment related toxicities) SIRT: 4/39 (10%) grade 3 or more treatment-related toxicities (abdominal pain)
Swinburne [54]	Retrospective study	34, but 5 patients were excluded without histological confirmation of ICC		11/29 (37.9%)	Surgery: 7 patients Chemotherapy: 15 patients TACE: 1 patient EBRT: 2 patients.		6.8–4.1	Median survival: 9.1 months (95% CI: 1.7–16.4)	No major toxicity was observed
Jia [55]	Retrospective study	24				1.6 ± 0.4 GBq.		9.0 months (5.6–12.4)	Grade 3 toxicities were observed in 20.8% patients (5/24) (abdominal pain and vomiting)
Soydal [56]	Retrospective study	16		3/16 (19%)	Chemotherapy: 9/16 patients TACE: 1 patient Surgery: 2 patients	Mean 1.7 ± 0.1 GBq		The median overall survival time was calculated as 293 ± 70 days (154–431, 95% CI)	Data not known
Saxena [57]	Retrospective study	25	Bilobar: 80%	12/25 (48%)	Liver resection: 10/25 (40%) Chemotherapy: 18/25 (72%) Ablation: 2/25 (8%) TACE/SIRT: 2/25 (8%)	1.76 GBq (SD = 0.33; range, 1.0–2.21 GBq		Median survival: 9.3 months	3/25 patients (12%) developed grade III albumin, alkaline phosphatase and bilirubin toxicity. 1 patient (4%) developed a duodenal ulcer.
Manceau [58]	Retrospective study Concomitantly with chemotherapy	35	Bilobar: 71.4%			2.6 ± 1.4GBq	Mean: 7.85 ± 3.47	Median OS was 28.6 months (95% CI: 21.8 to ∞)	6/35 (17%) patients presented hepatic dysfunction 1 patient presented with grade 3 cholecystitis.

Outside this clinical trial, the majority of patients receiving SIRT were pre-treated (up to 70%), with surgery, TACE, chemotherapy or external beam radiation [43,54]. In most series, patients had extrahepatic metastatic disease (from 1/4 to 1/3 of patients) [42,44,45,48,51,54,56,57]. 

The aim of SIRT is to deliver a sufficient dose to the tumor in order to induce a local response while sparing healthy liver tissue. A personalized dosimetric approach is important and necessary to select the best candidates for SIRT [58]. Bilobar disease does not permit the delivery of the optimal dose to the tumor in preserving a sufficient hepatic reserve.

A high percentage of liver involvement is frequently associated with poor outcomes [44,59]; therefore, patients with a tumor burden of >50% should not be treated with SIRT. Indeed, the reported median overall survival for patients with a tumor burden > 50% ranged from 1 to 6 months [44,49]. Other prognostic factors were associated with poor outcomes, like previous biliary stenting, a primary location different from iCC, an Eastern Cooperative Oncology Group performance status (ECOG PS) ≥ 1 and progressive tumors after the 1st line of chemotherapy [49].

The most frequent complications occurred in the first 30 days and included gastritis, gastrointestinal ulcerations, radiation cholecystitis and radioembolisation-induced liver disease (REILD). Grade 3 or 4 adverse-related events occurred in less than 10% in most of the series [42,45,48,50,51,59]. This incidence was increased when chemotherapy was given alongside SIRT. In the phase II trial MISPHEC [36], 71% of patients experienced grade 3 or 4 adverse events; hematologic toxicity was prevalent and was probably due to the chemotherapy. Two treatment-related deaths were reported in the 30 days following SIRT for patients with an extrahepatic disease and a PS of 2 [57]. Permanent liver toxicity was associated with the presence of cirrhosis and Child–Pugh score (A6 and B7) [58].

In conclusion, for well-selected patients with unilobar liver involvement, good liver function and a good performance status (PS 0 or 1), SIRT provides promising clinical outcomes. Its preferred indication is in the pre-surgical treatment setting in order to enable surgery in patients with initially unresectable cholangiocarcinoma (Figure 1C). Combination chemotherapy (gemcitabine-cisplatin) with SIRT led to a conversion rate (to resection) of 20% in a well-selected population [36,49].

#### 2.3.2. Transarterial Chemoembolization (TACE)

Several groups have proposed chemoembolization as an LRT for patients with unresectable cholangiocarcinoma. Most of the studies reported to date were retrospective (Table 4), and the number of patients in each of the studies was small. There was no standardization regarding the chemotherapy drugs used, so data were heterogeneous. In some cases, systemic chemotherapy was also administered concomitantly [53,60,61,62].

We found only one randomized clinical trial (n = 48) that evaluated TACE alongside systemic chemotherapy versus chemotherapy alone [62], with an improvement of PFS of 20 months in the combination arm. The phase 2 trial by Martin et al. [62] reported an improved median overall survival (mOS) for patients in favor of GEMCIS + TACE versus TACE alone (mOS: (33.7 (95% CI 13.5–54.5) months versus 12.6 (95% CI 8.7–33.4) months, *p* = 0.048)). In a carefully selected population, it could enhance median overall survival (mOS) compared to supportive care [63].

Patients with Child–Pugh B liver function, a PS ≥ 1, a hypovascular tumor, a tumor size larger than 5 cm or a multifocal tumor have the poorest clinical outcomes [23,64,65].

**Table 4 cancers-15-04217-t004:** Studies that evaluated TACE in iCC.

Authors	Retrospective or Prospective Study	Chemotherapy	Patients (n)	Median Number of Tumor	Median Tumor Size (cm)	Localisation	Mean Number of Sessions/Patients	Extrahepatic Disease (%) of Patients	Efficacy	Grade 3–5 Toxicities
One arm										
Zhou [66]	Retrospective study	DEB-TACE Epirubicin	88			Bilateral 33%		50%	ORR = 65.9% mOS 9 months	No grade 3–4 toxicity was observed
Luo [67]	Prospective trial	DEB TACE	37		5.7 (3–8.3)	Bilobar 27% Multifocal 67.6%			ORR = 66.7% after DEB-TACE treatment, mean OS of iCC patients was 376 days (95% CI: 341–412 days)	Data not known
Goerg [68]	Retrospective study	100 mg cisplatin (CDDP), 50 mg doxorubicin and 10 mg mitomycin C	18			Bilobar 52%	3.4		mOS: 13.3 months (0.95-CI 8.9–17.7 months ORR = 61%	1 severe toxicity 2 patient deaths due to liver abscess and sudden cardiac arrest.
Aliberti [69]	Prospective cohort	Doxorubicin DEBDOX and LIFDOX	127					0	Median OS of the LIFDOX group was 14.53 (95% confidence interval = 9.17–15.23) months.	DEBDOX: grade 3 toxicities were nausea/vomiting (24%) and fever (7%) LIFDOX: grade 3 toxicities were pain (7%). No grade 4 toxicity
Hyder [65]	Retrospective study	Gemcitabine + cisplatin/ Cisplatin + doxorubicin + mitomycin Gemcitabine alone Cisplatin alone	198 TACE 128 DEB TACE 11 Embolization 13 RE 46						mOS 13.2 months (95% CI 10.8–15.8)	16.8% patients developed a major complication (acute renal and hepatic failure, pulmonary embolism and liver abscess).
Vogl [64]	Retrospective study	Mitomycin C Gemcitabine Mitomycin C and Gemcitabine Mitomycin C, Gemcitabine and Cisplatin	115			Bilobar 77.4% 59.6% multiple (>5)	Mean of 7.1 (range, 3–30)	0	mOS: 13 months	No major complication was reported.
Kiefer [70]	Retrospective study	Mitomycin-C, doxorubicin and cisplatinum	62				Mean, 2.0; range, 1–4	19%	Median survival from time of first chemoembolization was 15 months	Major complications occurred following 5 of the 165 procedures (3%) (pulmonary edema and elevated cardiac enzymes post procedure, a pulmonary infarct, postembolization syndrome, acute renal failure and dehydration post procedure.)
Schiffman [60]	Retrospective study	Irinotecan doxorubicin	24		11.5 (4–33.3)	Median number of liver lesions was 3 but ranged from 1 to 25 lesions		10%	ORR: 79% mOS: 17.5 months	4 events of grade 3–5 toxicities were observed (hepatorenal syndrome led to death, sepsis from a port infection, hepatic insufficiency)
Shitara [71]	Prospective cohort	Mitomycin			7.8 (range 3.0–16.0)	Number of tumors 3 (1 × 10^10^)			ORR: 50.0% mOS: 4.1 months	5 patients (25%) presented grade 3–4 toxicities (gastroduodenal ulcer, epigastralgia)
Gusani [72]	Retrospective study	GEMZAR CDDP OXALIPLATIN	42		9.8 cm (range 1.3–17.0)		Median of 3.5 TACE treatments per patient (range 1–16)	19%	Median overall survival from the date of first TACE treatment was 9.1 months	2 patients presented grade 4 toxicities (acute myocardial infarction and hepatic abscess)
Comparative arm										
Martin [62]	Prospective trial	IRINOTECAN and GEMCIS (GEMZAR and CISPLATIN) concomitant IV prospective, multicenter, open-label, randomized phase II study	48 patients: 24 treated with GEMCIS and DEBIRI and 22 with GEMCIS alone						Median OS: 33.7 months (95% CI 13.5–54.5)	Data not known
Ge [23]	Retrospective study	Epirubicine + 5FU Comparison with MWA							mOS 26.9 months (6.6–44.2)	Data not known
Wright [61]	Retrospective study	Surgery vs. IAT GEMCIS (63%) GEMZAR (19.5%) IRINOTECAN (4.9%) CDDP-DOXORUBICIN-MITOMYCIN C 2.1%	59 patients underwent intra-arterial treatment (IAT) (41 = TACE, 16 = HAIC and 2 = SIRT ) vs. 57 patients who benefited from surgery	IAT: 5 (2–50) HAIC: 7 (2–50) TACE: 4 (2–27)	10.6 (3.3–25.3) HAIC 9.4 (4.1–19.2) TACE 11.0 (3.3–25.3)	Bilobar: 88% HAIC 81.3% TACE 90.2%	Median of 3 for the whole cohort of IAT (1–15)		mOS for IAT: 16 months (95% CI 13.3–18.7, *p* = 0.627) For TACE: mOS: 15 months (95% CI 11.4–18.6) HAIC pump = 39 months (95% CI 32.7–51.3)	Data not known
Akinwande [53]	Retrospective study	SIRT: 25 TACE: 15 DOXORUBICIN							ORR 6%	TACE: 3/33 (9%) grade 3 or more treatment-related toxicities (fever and abdominal pain) SIRT: 4/39 (10%) grade 3 or more treatment-related toxicities (abdominal pain)
Scheuermann [73]	Retrospective study	Surgery vs. TACE	273 130 surgery 32 TACE 111 palliative		8.7 (2.0–18.0)	Unilobar 13/32	Median: 3 (range: 1–18 sessions)		Median survival of TACE patients: 11 months	1 liver dysfunction (ascites) and 2 vascular complications (dissection or occlusion of the hepatic artery)
Park [63]	Retrospective study	TACE vs. palliative treatment CDDP	155 72 TACE 83 palliative		Mean 8.1 ± 3.4	Bilobar 37/72 Multiple or diffuse 41/72	2.5 per patient (range: 1–17 sessions)		12.2 months (95% CI 9.8–14.6)	11 grade 3 hematological toxicities occurred in 9 patients (13%, 9/72), and 25 grade 3 non-hematological toxicities occurred in 17 patients (24%, 17/72) (elevation of liver enzymes, pain and nausea)

TACE procedures were generally well tolerated with predominant gastro-intestinal (nausea, vomiting and abdominal pain), general (fewer) and hepatic (elevation of transaminases) toxicities and grade 1–2 adverse effects (AE) [66]. In most cases, grade 3–4 AEs were in around 10% of patients [53,65] but could reach 1/4 of patients [63]; principally, GI toxicities or hematological toxicities were seen, but renal failure was as well. If chemotherapy was also given, the rate could reach 1/3 [62], but toxicities were predominantly due to intra-venous chemotherapy. Grade 4 or 5 cardiac toxicities were rare but reported in different series [68,72]. In a series of 18 patients, two adverse fatal events were reported: one due to myocardial infarction and one due to sepsis from biliary abscesses [68]. One hepatorenal death was due to the procedure [60].

In conclusion, for multifocal lesions with a tumor burden < 75 or 50% and in patients with PS ≤ 2 and good liver function (Child-Pugh A5–6 or B7), chemoembolization seems to permit good control of local tumor growth (Figure 1D).

#### 2.3.3. Hepatic Arterial Infusion Chemotherapy (HAIC)

HAIC has been less studied than TACE or SIRT. One of the reasons is probably the necessity for a catheter implementation to access the hepatic artery, sometimes with a placement of a pump, which requires specialist expertise and more challenging logistics than other locoregional options. Some of the series identified were prospective clinical trials, but most studies still had a limited number of patients (Table 5). In some series or clinical trials, HAIC was given with concomitant systemic chemotherapy. In contrast to some of the earlier techniques, large tumors and multifocal disease can be treated by HAIC.

Two phase I and I/II clinical trials evaluated HAIC prospectively. In the phase I/II trial, 29 patients were treated with hepatic arterial infusion using gemcitabine without systemic treatment. This treatment was well tolerated but not as effective as expected, with a tumor response rate of 7% [74]. In the phase II study, 38 patients received concomitantly HAIC floxuridine and systemic gemcitabine and oxaliplatine. They observed an encouraging response rate of 58%. The median OS was 25.0 months (95% CI, 20.6-not reached), and the median PFS was 11.8 months (one-sided 90% CI, 11.1) [75]. 

Frequent toxicities observed with HAIC were predominantly elevated liver enzymes and gastro-intestinal toxicities (abdominal pain and nausea) or hematological toxicities, but they were also caused by the catheter placement: extravasation, obstruction or damage of the catheter [74,75].

The most frequent grade 4 relative toxicities that occurred in 4–10% of patients [75,76,77] were infection in the pump pocket, artery aneurysms or portal hypertension.

In conclusion, small prospective trials support the activity of HAIC in iCC. The technique may be most appropriate in the setting of large and/or multifocal tumors. Fewer centers are experienced in the technique (Figure 1E).

**Table 5 cancers-15-04217-t005:** Studies that evaluated HAIC in iCC.

Authors	Retrospective or Prospective Study	Chemotherapy	Patients (n)	Tumor Size Median (cm)	Extrahepatic Disease (%) of Patients	Number of Sessions per Patient	Efficacy	Grade 3–5 Treatment Related Toxicities
Cercek [75]	Phase 2 clinical trial	HAIC floxuridine and systemic gemcitabine and oxaliplatin	42 included and 38 treated	8.3 (1.7–24.8) Bilobar: 66%	18%		The median OS was 25.0 months (95% CI, 20.6-not reached)	The most common grade 3 and 4 adverse events were related to elevated liver enzymes (5% grade 4 elevated bilirubin level, 5% grade 4 elevated AST (aspartate aminotransferase), and 5% grade 4 elevated ALT (alanine aminotransferase)). No grade 4 non-biological toxicities were observed.
Marquardt [78]	Retrospective study	Melphalan	15			Range: 1–5	Median OS was 26.9 months from initial diagnosis and 7.6 months from first PHP	13 patients (50%) presented grade 3–5 toxicities (hematological, pneumonia, acute renal failure, ascites, bleeding, oedema, multi-organ failure, otitis, pseudoaneurysm and stroke)
Higaki [76]	Retrospective study	CDDP + oral S1	12	Multiple (35.7%)			Median survival time = 10.1 months (range, 3.6–23.2)	1 patient (4.5%) experienced a grade 3 toxicity (anemia)
Konstantinidis [79]	Retrospective study	5FU pump concomitantly with chemotherapy IV	167	8.5 cm (range: 1.5–16.4 cm) multifocal (63.5%)			mOS: 30.8 months	Data not known
Massani [80]	Retrospective study		11		0		mOS: 17.6 months (6–40)	4 patients experienced a major complication (hepatic decompensation and hand–foot syndrome)
Kasai [77]	Retrospective study	Fluorouracil and oxaliplatin after placement of an + HAIC pump + PEG-IFNa-2b SC	20		5%	Mean: 2 cycles (range: 1–8 cycles)	ORR 50% Median survival time: 14.6 months (95 % CI 5.5–16.8)	6 patients experienced grade 3 hematological toxicity
Ghiringhelli [81]	Retrospective study	HAIC GEMOX	12	Multifocal 5/62			ORR 66.6% (95% CI 29–100%) Median OS: 20.3 months (95% CI 13.2–49.7)	7 grade 3–4 hematological adverse events and 6 grade 3–4 non-hematological adverse events were reported (oxaliplatin-related peripheral neuropathy, infection and oxaliplatin-allergy)
Inaba [74]	Phase I/II clinical trial	HAIC GEMZAR	11		9%			The incidence of adverse events of grade 3–4 was 20% neutropenia, 22% elevated liver enzymes, 4% nausea and 4% fatigue.
Mambrini [82]	Retrospective study	EPIRUBICIN AND CDDP + CAPECITABINE oral	20				OS 18 months	One grade 5 toxicity (diarrhea) and one grade 3 toxicity (vomiting).
Vogl [83]	Retrospective study	GEMZAR	24				OS 20.2 months	1 severe adverse event occurred (allergic or toxic lung edema)
Cantore [84]	Retrospective study	EPIRUBICIN + CDPP 5FU IV	30			Median 4 (2–8)	ORR 40% mOS 13.2 months	Grade 3 toxicity observed in 11 of 30 patients (37%) (hematological toxicity, stomatitis, nausea, diarrhea, alopecia)
Tanaka [85]	Retrospective study	Epirubicin and cisplatin 5FU	11	Mean tumor size: 7.0 ± 2.6 cm (range: 3.8–13.5)	4%			One severe cholangitis was observed

#### 2.3.4. Comparisons of the Different Intra-Arterial Therapies

For LRT, IAT could be used in a palliative setting or in a neo adjuvant setting in order to downsize initially unresectable tumors to resection.

There are no randomized clinical trials comparing all the LRT modalities to help clinicians in decision making; nevertheless, based on the available retrospective series and clinical trials, we have derived some recommendations that may be applicable in daily practice.

Patients with extrahepatic disease were included in many of the studies, but these patients had unfavorable outcomes; therefore, these patients had to be treated with a systemic treatment. Cirrhotic patients could benefit from LRT but only with preserved liver function and a Child-Pugh score of A.

### 2.4. Existing Guidelines

For unresectable iCC, LRTs are included in the proposed treatment options; however, there are no clear recommendations for choosing between the different modalities, and there are no specified criteria to help clinicians choose the best option [86].

#### 2.4.1. NCCN (National Comprehensive Cancer Network)

For unresectable iCC, NCCN recommendations [87] propose LRTs as an option. Concomitantly to radiotherapy, the network proposed a chemotherapy with fluoropyrimidine. There is no specific chapter for intra-arterial therapy in iCC, but it proposed the same modalities as in HCC.

#### 2.4.2. ESMO (European Society for Medical Oncology)

In the curative setting, the ESMO guidelines only recommend surgery [88]. For unresectable non-metastatic cholangiocarcinoma, IAT could be proposed as an option and in combination with systemic chemotherapy, but there are no details on which technique is preferred or which criteria would be optimal for a patient or tumor.

Radiotherapy is included as an option for localized disease, and for unresectable iCC, SIRT is also an acceptable option.

## 3. Which Treatment for Which Patient?

The spectrum of liver-limited iCC varies greatly, and the applicability of the different LRTs differs, as their efficacy and safety profile differ between different situations. First and foremost, it should be stated that the level of evidence of the different LRT options is currently low and that they should not replace the standard of care, namely surgery, for resectable tumors and systemic chemotherapy (±immunotherapy if available [89]) for unresectable tumors. However, we believe that there is a potential benefit from these therapies, either in the case of the non-feasibility of the standard of care treatment or in conjunction with this treatment. Factors that will influence the choice of the LRT are the following: 

Patient-related factors (age, comorbidities, concomitant medication, etc.);

Background liver-related factors (cirrhosis);

Disease-related factors (proximity to vessels (blood and/or biliary)), the maximal size of the lesions, number of lesions and unilobar vs. bilobar disease);

Local expertise;

A clinical trial option.

Cirrhosis and comorbidities could render surgery more difficult but might be a good indication for RFA, MWA or EBRT. Vessel invasion or proximity might contra-indicate surgery or RFA and MWA but might be accessible for EBRT. Some LRTs show better results in small lesions (<3 cm for RFA or MWA) or intermediate-size lesions (<5 to 7 cm for EBRT or TACE), while others have no real limit (SIRT and HAIC). A low number of lesions is preferred for RFA, MWA, EBRT and SIRT, while an intermediate number is better for TACE, but the number is not a limitation for HAIC. Bilobar treatment is associated with increased toxicity with SIRT.

Some clinical settings are thus “ideal” for each treatment modality (Figure 2). A small lesion in a cirrhotic or previously operated liver is an excellent setting for RFA or MWA (Figure 2A). A similarly small, unique lesion close to a vascular structure is a candidate for EBRT (Figure 2B). A larger lesion, with unilobar diffusion, would be a preferred candidate for SIRT (Figure 2C). A diffuse disease could be only accessible to HAIC (Figure 2D).

However, in the real-life setting, the discussion is more precisely represented as a continuum, with various factors influencing the discussion (Figure 3). Each patient case should be discussed individually, and a precise multiparametric evaluation should be performed, along with an analysis of the clinical parameters (cirrhosis, performance status and previous treatment received), as well as the imaging parameters (liver CT scan, MRI and extrahepatic spread evaluation). The role of an expert multidisciplinary team (MDT)’s discussion, with the involvement of every specialist (surgeon, interventional radiologist, radiation oncologist, nuclear medicine physician and medical oncologist) is obviously of paramount importance. In some settings where expertise is lacking locally, referral to an expert center should be discussed to assess whether an LRT could be provided to a specific patient.

## 4. Conclusions

The role of different modalities of LRT for iCC remains unclear due to a lack of randomized comparative clinical trials and many studies being retrospective and involving a low number of patients. LRT could be proposed after discussion in a multidisciplinary board of experts, always bearing in mind that the standard of care treatments remain surgery and systemic chemotherapy. Due to the lack of relevant proof, performing adequate prospective clinical trials is of paramount importance for the future definition of the adequate positioning of LRT in iCC.

## Figures and Tables

**Figure 1 cancers-15-04217-f001:**
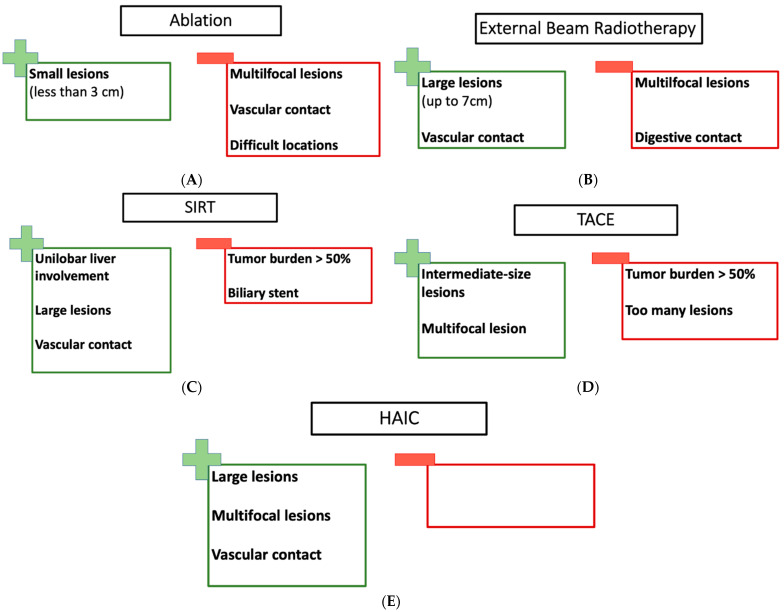
(**A**) The pros and cons of RFA and MWA; (**B**) the pros and cons of EBRT; (**C**) the pros and cons of SIRT; (**D**) the pros and cons of TACE; (**E**) the pros and cons of HAIC.

**Figure 2 cancers-15-04217-f002:**
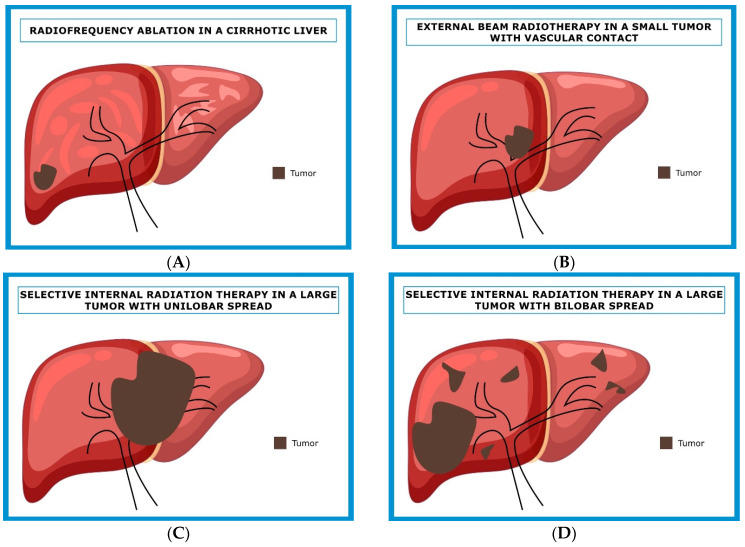
(**A**) Potential ideal candidate for an ablation; (**B**) potential ideal candidate for external beam radiotherapy; (**C**) potential ideal candidate for SIRT; (**D**) potential ideal candidate for HIAC.

**Figure 3 cancers-15-04217-f003:**
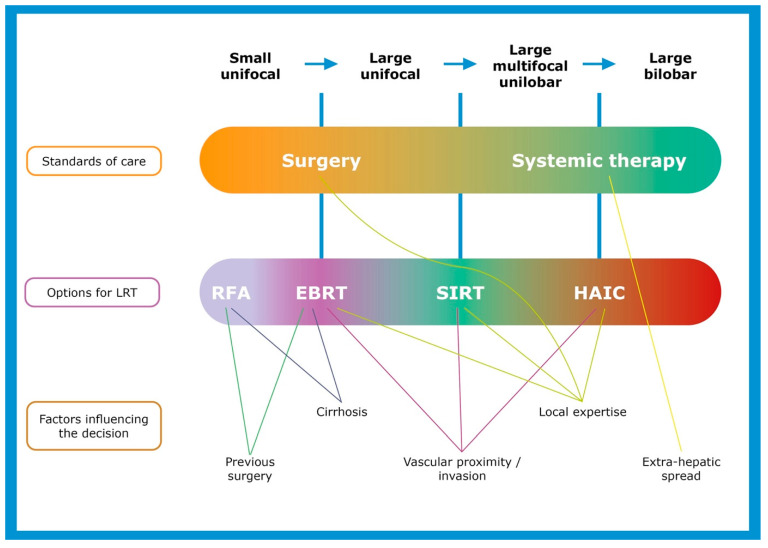
Proposed continuum of care for selection of potential locoregional treatment for liver-only intrahepatic cholangiocarcinoma.

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
