# Peer review of "Locoregional Treatment in Intrahepatic Cholangiocarcinoma: Which Treatment for Which Patient?"

_cancers, 2023, doi:10.3390/cancers15174217_

Round 1
Reviewer 1 Report
This review is basically well written with a large endeavor of searching vast literature. Nevertheless, lacking the 'key message' is the drawback; of which means, although the authors gathered large information, readers may not reach clinically useful information.
Anyway, it is agreeable that this review is quite informative. I have few minor queries.
1) ablation (get rid of, in terms of origin) may not means RFA or MWA. the term ablation has wider meaning. You may use term RFA or MWA specifically.
2) author said 'However, prospective randomized trials are required to better position this treatment.' in EBRT section. however, none of the LRT modalities are supported by good evidences from RCTs. Therefore the sentence should be removed and rather description about useful indication of radiotherapy (for example, vascular contacting tumors, larger tumors than those suitable for RFA) should be more described.
3) try to merge many figure 1s, for example in a flowchart system, so there can be some clinically useful information (although it can be authors' opinion).
4) Figure 2c: is this picture is really 'unilobar' spread?
Figure 2b: is this picture for 'streotactic radiothearpy' or 'external radiothearpy'? they are not same term. SRT is part of EBRT.
Author Response
We thank the reviewer for its suggestion, you could find attached our reply.
Sincerely

Reviewer 2 Report
The publication reviews locoregional treatment strategies for intrahepatic cholangiocarcinoma. It is based on a prior publication (ref 7), adding description and interpretation of methods, in particular trying to define factors that make a patient well or less suitable for a specific method. Overall it is well written and structured, including Tables that list variuos studies for a specific method and Figures that explain important factors for a specifc method.
An additional column in the Tables considering important toxicities for the respective treatment techniques would be helpful.
Author Response
We thank the reviewer for its remark, as suggested, we add a column for grade 3 to 5 related treatment toxicities for each modality of treatment.
Sincerely